# Technologies Supporting Screening Oculomotor Problems: Challenges for Virtual Reality

**Are Dæhlen** *, **Ilona Heldal** and **Qasim Ali**

Department of Computer Science, Electrical Engineering and Mathematical Sciences, Western Norway University of Applied Sciences, 5096 Bergen, Norway; ilona.heldal@hvl.no (I.H.); qasim.ali@hvl.no (Q.A.)
* Correspondence: ard@hvl.no

**Abstract:** Oculomotor dysfunctions (OMDs) are problems relating to coordination and accuracy of eye movements for processing visual information. Eye-tracking (ET) technologies show great promise in the identification of OMDs. However, current computer technologies for vision screening are specialized devices with limited screen size and the inability to measure depth, while visual field and depth are important information for detecting OMDs. In this experimental study, we examine the possibilities of immersive virtual reality (VR) technologies compared with laptop technologies for increased user experiences, presence, immersiveness, and the use of serious games for identifying OMDs. The results present increased interest in VR-based screening, motivating users to focus better using VR applications free from outside distractions. These limitations currently include lower performance and confidence in results of identifying OMDs with the used HMDs. Using serious games for screening in VR is also estimated to have great potential for developing a more robust vision screening tool, especially for younger children.

**Keywords:** eye-tracking; head-mounted display; presence; immersiveness; oculomotor dysfunction





## 1. Introduction

Eye disorders are reasonably frequent among the population. Many common problems can be identified by visiting clinical experts, e.g., ophthalmologists, orthoptists, or some clinical specialists in vision, and corrected with eyeglasses or surgery. However, vision problems can occur even if the eyes seem normal, and the results from the usual vision tests (e.g., visual acuity assessment, refraction for eyeglass prescription, or examination of the anterior and posterior segments of the eye) do not show vision disorders [1,2]. Some people may have problems processing visual information, addressed in this paper as functional vision problems (FVPs). These problems, also called functional visual disorders or functional vision impairment, refer to disturbances that cannot be explained by structural or physiological abnormalities of the eyes. Many of these conditions are characterized by a mismatch between the diagnosed eye health and the visual problems experienced by the individual. These sight disturbances hinder one from correctly estimating distances and problems, e.g., for reading, experiencing blurry vision, headache, or balance problems. FVPs are common, especially in stroke patients (92%) [3–5] or in adults suffering from brain injury (60–85%) [6]. Not diagnosing FVPs can have negative consequences, especially for children who do not necessarily realize and report their problems and are not given the usual vision testing at the ages of 5–7 [7].

Oculomotor dysfunction (OMD) is an FVP related to problematic coordination between the left and right eye. Approximately 17–30% of children with vision problems have problems related to FVPs such as OMD [8]. These problems can lead to more severe vision disorders if not treated correctly; however, many children do not know that they have OMD [8]. FVPs are a societal problem that cannot be solved with current resources due to the limited number and capacity of vision professionals [7,9].

Utilizing eye-tracking (ET) technologies shows great promise in the identification of OMD [10]. These ETs are integrated or attached to laptop systems and, based on following a person's basic eye movements for a period of time, help professionals assess if the person has or does not have OMD-related vision problems. Due to the limited screen size and the inability to measure depth accurately, essential issues for a complete vision screening, these solutions have inherited limitations. Measuring basic eye movements helps professionals understand how a person can focus on objects (measuring fixations), follow objects with their eyes (measuring smooth pursuits), or how their eyes jump from one object to another. Since the movements from both eyes can be measured separately, and how the eyes are coordinated and process information on visual stimuli can also be measured, this can provide an effective solution for examining FVPs related to OMDs. There are already validated solutions on the market offering ET and laptop-based measures to professionals engaged in screening FVPs [11]. Supporting screening via technology is essential since vision screening should be based on objective measures and take less time. A complete vision screening, including screening for OMDs, takes more than one hour [11], and there are few professionals who are educated to perform this.

Immersive VR technologies, e.g., HMDs, Hololenses, and VR rooms, allow users to interact directly with surrounding computer-generated 3D graphics, with possibilities of achieving higher user experiences and increased presence [12,13]. Medical VR applications take advantage of the technology's ability to elicit emotional responses and convey spatial information [14,15]. Since VR equipment allows experiencing a larger field of view (FOV) and depth, the hypothesis behind this work is that VR can add to future vision screening batteries. Until now, we are not aware of research or practice utilizing VR equipment for vision screening. This may be evident due to the main limitations of VR to exactly measure ocular movements and positions, e.g., handling "binocular disparity is a critical stimulus to vergence, which is a critical depth cue" making sure the eyes "are always focused on a single depth", which implicates loss of focusing and is accommodated in a current review [16] considering ET in VR. However, this review is positive for utilizing larger FOV, by a foveating view or for some specific vision testing, e.g., those requiring contrast sensitivity [16]. Another review, also considering AR technologies, highlights limitations such as including unsatisfactory accuracy, weak validation, and hardware limitations, but also discusses benefits from high user experiences [17]. However, not many studies examine these experiences in detail.

By differentiating the support of vision problems into two main groups, we can consider that the first group includes identifying the problems and the second includes supporting already identified problems, if possible. This paper focuses on VR-supported identification or screening, which are performed in many cases today by professionals. There are only a few studies illustrating the promising use of such technologies, such as prototype tools for identifying vision-related disabilities in patients with glaucoma (e.g., [18,19]). Prototype applications for functional vision assessment have also been investigated; however, these tools focus on patients with low vision and were only tested in lab settings (e.g., [20,21]).

VR is also a technology that allows supporting vision training [8], which can help train, e.g., stroke patients [22] or the vision of children with OMD [23]. Since many FVPs, and in particular OMDs, can be helped by training [24], to see the effects of the training, this should also incorporate screening possibilities [20,21]. According to our current knowledge, there is no VR application for screening OMD and no VR eye-training application incorporating the possibilities of screening, except an experimental study implementing a laptop-based application to screen FVPs and developing an immersive virtual reality (VR)-based prototype and evaluating its usability [25], motivating this research.

This paper takes a step forward and focuses on the importance of VR experiences for OMD screening and the current challenges through experiencing presence during vision screening. Therefore, the paper takes an overall approach on technology used for OMD

screening which is not limiting the aim to information and software. The overall goal is to highlight user opinions on utilizing VR for screening and training of vision problems. This focuses on their experience with ET calibration techniques, experiencing of presence, comparison between a laptop and VR-based solution, and feedback from open-ended questions and interviews which provide additional justification to questionnaire answers.

While the value of having OMD, and later FVP screening, in VR is clear, the way to achieve it is long and depends on several aspects of technology, context, application, and users [26]. Only the technical setting includes a combination of computer or VR technologies including ETs, and has to be further developed to find guidelines to determine how the different parts of the technology influences accuracy for eye measurements [27]. Experiencing presence for VR is important and should also affect VR-based screening.

The structure of this paper is as follows. We present the related literature from VR-based and laptop-based screening in Section 2. Section 3 presents a screening tool (C&Look) that is shown on a laptop and imported into VR. The study design, including data collection methods, test approach, and a brief introduction to analysis methods, are described in Section 4. Section 5 presents an analysis of answers to questionnaires and interviews. The context for these results are discussed in Section 6, along with a presentation of current limitations and future work for this project. Section 7 provides a conclusion to this paper, presenting an overview of the findings from this study.

## 2. Literature Background

VR provides a surrounding experience by simulating a real-world environment with the help of technologies and users can be surrounded by 3D projection in a room. For example, in a head-mounted display (HMD), when the user is wearing special glasses, allowing them to see 3D projections around themselves [28]. Immersion, as defined in the literature, refers to the characteristics of technology that allow experiencing this 3D environment in space, not only on a 2D surface. Accordingly, an HMD is an immersive technology and a laptop is not. Presence refers to experiencing being physically present in a computer-generated application, and the interaction in it can be as believable as the interaction in the non-mediated conditions [12].

Immersive VR has gained significant attention from researchers due to allowing a larger field of view than a laptop, enabling more natural interaction, for example, with the hand, head, or body tracker with a computer-generated environment and built-in eye tracker for gaze recording. For enabling high presence or distracting users from painful or boring situations, VR is appreciated in various fields, from experiencing new architecture (e.g., [29]), training for emergency (e.g., [30]), or education (e.g., [31]). A current review examining the production of studies focusing on immersive VR in medicine questioned and enhanced this popularity by the large number (2700) of published studies, only in the last year in PubMed (e.g., [14]). Medicine utilizes VR technologies to train to be prepared for surgery (e.g., [32]), pain management (e.g., [33]), anatomical education (e.g., [34]), or the treatment of psychiatric disorders (e.g., [35]).

In recent years, VR has integrated ET technologies and emerged in the field of vision science, integrating built-in eye trackers into HMDs [16]. Therefore, today, VR has the potential to be an effective tool in complementing the treatment of a variety of vision disorders requiring ET technologies for identification or treatment, e.g., treating amblyopia [36] and convergence insufficiency [37]. VR and augmented reality (AR) are used for treating strabismus [38], amblyopia, and retinal diseases [17].

One of the most important eye problems to identify is amblyopia, or lazy eye, caused by three main factors: unequal refractive powers in both eyes (anisometropia), misalignment of the eyes (strabismus), and visual axis obstruction (deprivation) [39]. These factors lead to reduced vision in one eye due to the brain favoring the other eye or receiving insufficient visual input. Black et al. [40] performed a clinical test to measure amblyopia using virtual reality glasses where the amblyopic eye is exposed to stimuli with high contrast, while the

stimuli shown to the non-amblyopic eye have varying contrast levels. Patients engage in a signal/noise task, enabling precise evaluation of excitatory binocular interactions.

Several research studies have investigated the use of VR with eye trackers to detect ocular deviation angles in strabismus. This approach offers advantages over traditional methods used for measuring ocular deviation, including the Krimsky test, the alternative prism cover test (APCT), and the simultaneous prism cover test [41]. Economides et al. [42] investigated the use of VR and ET in strabismus patients with ocular deviations ranging from 4.4° to 22.4°. Strabismus severity is determined by the magnitude of ocular deviation, which can be quantified using numerical values. These numerical measures serve to express the extent of misalignment in strabismic individuals [43]. The findings from this study showed that the fixating eye of patients with strabismus exhibited greater variability in position compared to the fixating eye of individuals without strabismus.

Laptop technologies, in general, and for a longer time showed promise in complementing the treatment of amblyopia, strabismus, binocular vision disorders, and visual field deficits [44,45]. However, although developing associated algorithms and analyzing gaze measurements from ET data for fixations and saccades are available, more exact measurements are needed for better confidence in the results both for laptops and VR [10]. Experiencing presence in the environments is crucial, as an example, for feeling agency [46] and, therefore, our hypothesis is that evaluating user experiences, presence, and the role of serious games can help forward research about these technologies.

VR systems do not aim to reproduce an experience as realistic as in films or fiction; the experience and presence in the environment, and knowing how to react to the events, are important. Working with the technology, where the technology itself is hidden and goes away for the good of the application, is significant for increased user engagement, motivation, and enjoyment [28]. Since experiencing presence can be considered an added value for VR technologies, many tests aims to collect measurements about presence. These tests can be performed by addressing user opinions, e.g., by observations, questionnaires, or interviews, but also by trying to make sense of a user's action in the environments, e.g., by sensing technologies such as ETs or EEGs and finding more objective measures for presence.

Given the high prevalence of vision problems in the general population, functional vision screening is important for early detection and timely treatment, which can significantly improve visual outcomes and quality of life [7,47,48]. The literature also shows that using serious games increases motivation for learning or performing tedious, repeated, or painful activities, for example [49,50].

Despite the growing body of research on the therapeutic applications of VR in vision rehabilitation, to our knowledge, there has been no exploration of how subjects are experiencing the presence of VR-based vision screening.

## 3. The C&Look Application

In order to illustrate the development of the VR application, first, the laptop-based application is presented. This application, according to other commercial applications such as RightEye and iMotions, are based on measuring basic eye movements together with information related to stimuli.

### 3.1. Developing a Laptop-Based Application

The C&Look program developed at HVL uses affordable eye-tracking technology to support existing detection methods for OMDs [3]. It utilize eye-tracker technologies that have to be started on a laptop, calibrated and used with a structural, systematically developed application, designed together with vision experts [51]. The use of eye trackers connected to a computer has some limitations such as screen size, e.g., peripheral vision, and head positioning, e.g., not allowing larger head movements during the screening. However, it provides new possibilities as the data gathered over time allows for the analysis of eye movements, and such difficulties were earlier only possible to be examined

by ocular examination from vision experts. Further, it has illustrated that the affordability of eye-tracking technologies can assist vision experts in detecting children's OMDs and how they can incorporate this technology into their screening procedures [11].

The C&Look program runs on a laptop with ET technologies. Vision experts can choose and adjust a number of tasks and define a task battery for screening, depending on the possible problem of the users and their characteristics (most often age, but also eventual disability, for example, stroke is influencing reading competence). After calibration, screening is performed. After the screening, the performance of each user (how they perform the task and how their eyes, the left and right separately, are synchronized, e.g., by fixation, saccade, smooth pursuit) can be analyzed separately. The screening battery includes graphical tasks with measured fixations, saccades, smooth pursuit, and other measurements from reading tasks [3]. Fixation, smooth pursuits, and saccades are major eye movements connected to how people can focus and follow objects. Fixation tasks consist of an object moving stepwise across the screen in a predetermined pattern; smooth pursuit tasks follow objects, while saccades are measured for an object jumping from one position to another. Reading tasks display any text to the user, depending on her age and interests, adjusted by the person who sets up the screening. The program records the eye movements while the text is being read. This application also boasts its calibration method and a comprehensive results screen with replays and graphs visualizing both eye positions during testing.

The analysis toolkit superimposes the user's eye movements, allowing the examination of collected measures, alignments, and together with information about the performance (Figure 1). The analysis can be performed for one task or just specific parts which can be associated with special measurements. Current issues with the application include strict user positioning for gaze data collection and the inability to measure aspects of vision related to depth.

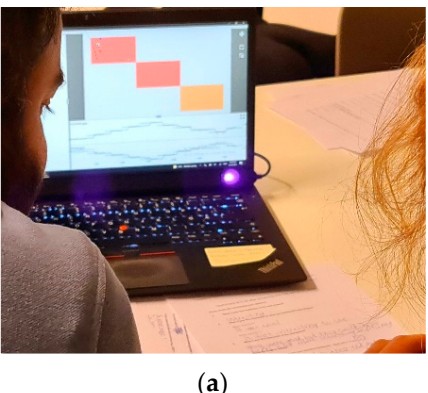 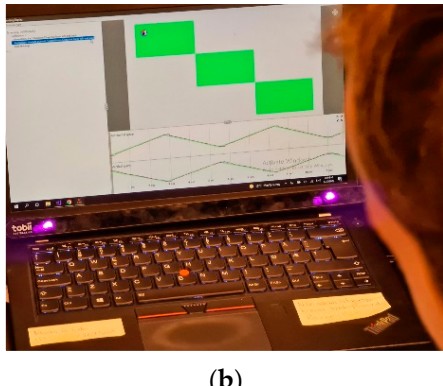

(**a**)          (**b**)

**Figure 1.** The analysis toolkit. Both pictures show the examination of the diagonal smooth pursuit eye movements for following a ball from the upper-left toward the lower-right corner. On the left (**a**), the pink–red (orange) color indicates bad (medium) eye alignment, contrary to a good alignment on the right picture (**b**).

### 3.2. Development of a VR Application

When transforming C&Look to VR, the first step was to convert the same tasks (fixation, smooth pursuit, reading) into a 3D environment. Both the fixation and smooth pursuit tasks follow the same principles when it comes to stimuli movement, although these objects are now 3D and move in a 3D space. The reading task in VR is almost identical, with text being displayed on a plan-screen in the virtual environment. The difference between this task from laptop and VR comes from the canvas positioning and being able to dynamically adjust the distance between the player and the text. Calibration in VR utilizes the Varjo SDKs' built-in calibration method, while results include a live replay with visualization of gaze points. The VR version currently lacks graph visualization of eye

positions during testing. Few implementation methods can be reused when developing VR, as the added third dimension changes object movement, gaze point visualization, and necessary calibration techniques. This led to this new application being built from the ground up within the Unity game engine, except for utilizing the same database.

To ensure that the application was scalable, data collection methods and task scenes were standardized. This allowed for quick experimental development of additional screening tasks based on manual vision screening procedures. The possibility to measure depth perception and hand–eye coordination was investigated; however, these tasks were later abandoned due to necessary complexity and inconsistent hardware. After these two failed attempts at implementing new screening tasks unique to VR, the development focus shifted to increasing the immersive aspects of the application through improved environments and more stable hand–controller interaction. Figure 2 shows a software architecture diagram of the developed VR vision screening application, giving an overview of the different components.

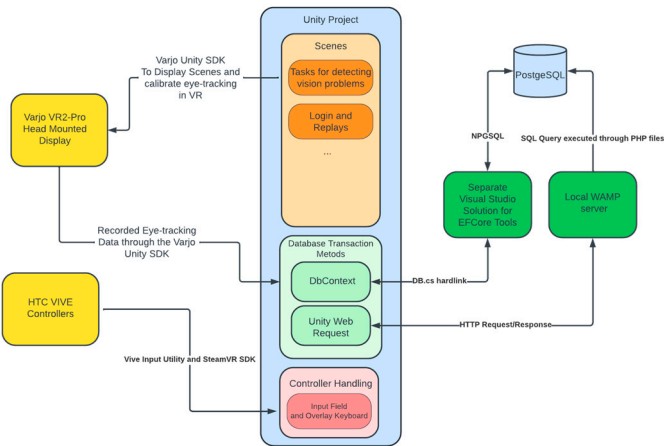

**Figure 2.** Software architecture diagram for the VR vision screening application.

To transfer C&Look into an immersive VR application using eye behavioral data, and to investigate how to complement manual vision screening and C&Look, the prospects and limitations of using a head-mounted display (HMD) were analyzed. The HMD used was a Varjo VR2-Pro, as it is one of the leading VR headsets with embedded ET together with hand controls (see Figure 3).

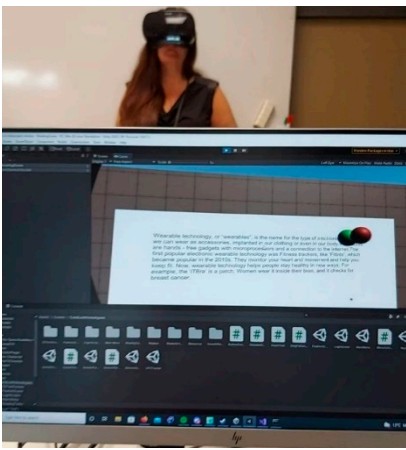

**Figure 3.** A vision expert testing a prototype of the VR-based screening application for reading.

The sampling rate of integrated ET in the Varjo HMD is 100 Hz. Varjo also includes hand-tracking software, presenting an opportunity to investigate hand–eye coordination.

This functionality is useful for recognizing FVPs, since problems with how the eye functions are highly correlated with other functions, e.g., balance or hearing [10]. An experimental task attempting to measure the correlation between hand and eye movements was developed for the VR solution; however, the accuracy of the hand-tracking was found to be insufficient for providing objective measurements [25].

While the two-dimensional version of C&Look provides high-quality data when testing for oculomotor problems (OMD), it lacks the possibility of capturing important tasks performed during a special performance that is included in manual vision screenings. The laptop system used for testing the original version of C&Look had a screen size of 14 inches and utilized a Tobii 4C mobile eye-tracker with a sampling rate of 90 Hz. The performance of the laptop-based C&Look application was investigated comparatively in a set including two applications for the same tests, one based on higher sampling rates and one with lower [52]. The applications were approximated to be similar, with testing performed by the involved vision teacher (see Figure 3).

### 4. Study Design

After implementing C&Look in VR, the program was tested several times by random students. Each participant tested both C&Look on a laptop and on VR, and quantitative data was collected from the participants. During this time, a test battery was developed to collect data anonymously concerning Norwegian ethical requirements. This test battery (presented in Appendix A) includes:

1. Information about this study (written and oral information).
2. Consent for participation.
3. Background information about the participant's familiarity with vision control and their familiarity with new technologies (ET, VR, and serious games).
4. Questionnaire about experiencing the laptop version of C&Look.
5. Questionnaire about experiencing the VR version of C&Look.
6. Comparison questionnaire for the laptop and the VR application.
7. Open-ended questions related to the feeling of presence and comparative elements of both applications.
8. ET calibration experience questionnaire.
9. Presence questionnaire for the laptop and the VR application.
10. Nine interview questions, with, after each, the possibility to discuss comparatively evaluating their presence and experiencing the technologies and serious games.

This experimental study was tested by seven subjects, one vision expert (a teacher in special education with competence in vision), and six other voluntary subjects in June 2022. Testing one subject lasted between 60 and 90 min. The first subject was one vision expert critically examining the test battery on the laptop and VR applications; the last one was adjusted after her comments. All participants, except one, had some vision difficulties, but all participants had earlier experienced vision testing at a vision specialist.

The test was performed in the following way: Introduction (using questionnaires 1–3), performing a randomly assigned vision screening (laptop or VR), and after each set, the subjects filled in a questionnaire (questionnaire 4 or 5). After the first application's vision screening and data collection was completed, the process was repeated for the remaining untested application. The evaluation of the user experience (UX) of the laptop and VR application was inspired by an overall UX questionnaire [53]. Each participant filled out the comparison questionnaire (6) after having tested and evaluated each application separately. This questionnaire began with open-ended questions related to the feeling of presence and comparative elements of both applications (6a). During the comparison questionnaire, participants were asked to rate their experience with calibration in each environment (6b). The activities during application testing involved two different calibrations on the laptop, one from Tobii and one developed locally, in this environment for more precise calibration. For the VR calibration, Varjo's legacy calibration method was used. These calibration methods were ranked on a scale from 1 to 7, where 1 indicates difficulties with calibration



and 7 relates to an easy calibration experience. For the comparison questionnaire between the experiences with the laptop and the VR application, a modified overall presence questionnaire (6c) was constructed, inspired by earlier work from Slater [54,55].

The testing session ended with semi-structured interviews about the comparative experiences, aligning the experiences to earlier familiarity with vision screening and using VR.

User experience and usability evaluation have been presented in an earlier study, showing similar high-quality results for both applications except for a few performance issues in VR [25]. The study also highlights barriers for the implementation of immersive technologies during development, including many limitations of the used HMD and low data quality, in addition to the need for interdisciplinary assistance from vision science domain experts when developing measurements for a vision screening suit for VR. For this study, this experience was based on vision expertise from an earlier study developing C&Look for laptops [3], as well as testing of an early prototype of the VR application by a vision expert. Another barrier for utilization of VR technologies for FVP screening comes from the translation of two-dimensional tasks into a 3D space. While representing the same predetermined movements in 3D as the ones used on the laptop 2D version of C&Look may seem like it provides equivalent measures, i.e., smooth diagonal movement of an object for smooth pursuit measurements, this task translation is only a hypothesis. The introduction of a third dimension may require new approaches for measuring eye movements, as different environments can affect eye-tracking data and user behavior. These aspects are important reporting guidelines for eye-tracking studies, which may also affect immersive virtual environments [56].

To analyze the data collected through questionnaires and interviews, different analysis methods were used. Calibration experience results have been plotted into a graph for better visualization (Section 5.1). Open-ended questions regarding comparative elements and presence were analyzed (Section 5.2). Presence questionnaire data were averaged separately for both applications, providing a mean sense of presence score for all tasks performed in different environments (Section 5.3). Interview answers have been compared to other results for user experience and presence, relating answers to each participant and their previous experience with vision screening/testing (Section 5.4).

## 5. Results

The results presented in this section include data and information from calibration experience results, code analysis of answers to open-ended questions regarding presence and experiences, task performance similarity results, and analysis of feedback received during interviews.

### 5.1. Calibration Experience Results

When collecting ET data from users, calibration of ET technologies is a mandatory step to ensure high-quality data collection. The laptop and VR screening applications use different calibration methods, with the laptop version using a custom-made calibration screen proposed by Eide and Watanabe [3], and VR utilizing Varjo's built-in legacy calibration mode. When asked to compare these methods in step 6a of the testing battery, ranking each calibration method on a scale from 1 to 7, feedback from test participants varied greatly. Scores for each calibration method per participant are shown in Figure 4. The average score for calibration on a laptop was 6, while the VR method scored an average of 5.85.

Both calibration methods produced similar averages, with the VR version falling slightly behind the laptop version. However, this difference was influenced mainly by Participant 5, who had difficulties calibrating in VR due to wearing glasses. On the other hand, Participant 7 also wore glasses but had no issues calibrating in either environment, giving both calibration methods a score of 7 when asked to rate their experience.

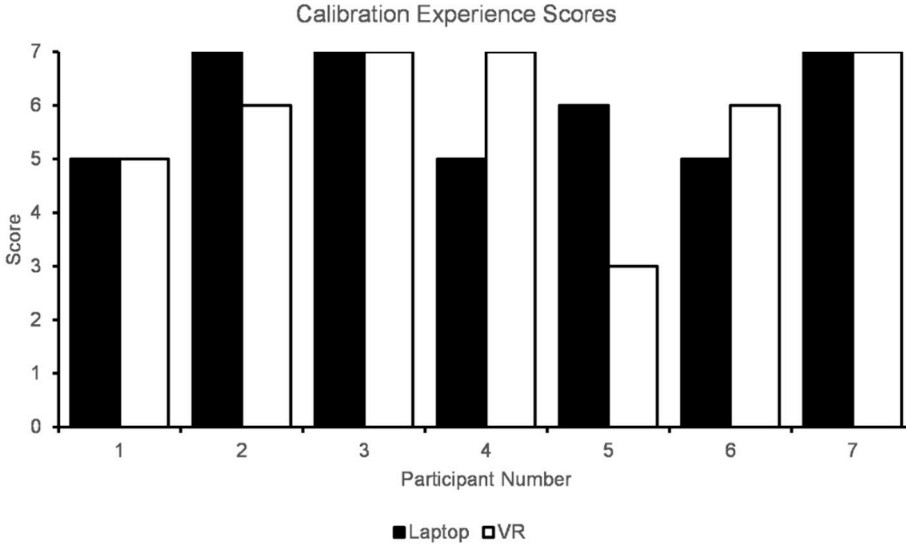

**Figure 4.** Calibration experience scores for each test participant, ranging from 1 to 7.

*5.2. Open-Ended Questionnaire Results*

Open-ended questionnaires were administered to participants in step 6a of the testing battery to compare different aspects of the two applications. The questionnaires were designed to obtain qualitative feedback from participants about their experiences with each application. The following are the questions:

- Q: Comparing with experiences while checking your eyes at a physical place, e.g., at an optician or a doctor's office, can you argue why (or why not) you would like to use a similar application on a laptop?
- Laptop:
- VR:
- Q: In which application did you find it easier to navigate? Laptop or VR? Why?
- Q: Compared to performing the tasks on a laptop, did the addition of depth in VR change your enjoyment/immersion? Why or why not?
- Q: Were there any features from either application that you felt were lacking from the other? If so, what?

To analyze participants' responses to the open-ended questions in step 6a, each answer was assigned a "code" based on its intention. Each response was given only one code, ensuring responses have the same weight. The following codes were used:

- "Laptop is easier to understand/use": indicated that participants found the laptop application easier to use, especially those with limited technological background or prior knowledge of the system.
- "Higher confidence in laptop results": indicated an interest in better data representation and collection for the VR environment, leading to higher confidence in the laptop application's results.
- "VR is more fun/exciting": highlighted the additional immersive elements that VR brings, with an emphasis on enjoyment.
- "VR helps with focus": included mentions of participants finding task performance easier or more motivating with fewer outside disturbances in VR.
- "VR is easier to navigate" and "Laptop is easier to navigate": described preferences for different user interfaces and navigation options.
- "VR performance issues": included responses that mentioned optimization issues in the VR application.
- "No Answer": contained answers that were non-existent or completely unrelated.

Figure 5 shows the number of occurrences for each code based on participant answers to each question. The numerical values in the graph represent the number of times an

open-ended question answer was given a specific code. As an example, the "VR helps with focus" code was given three responses.

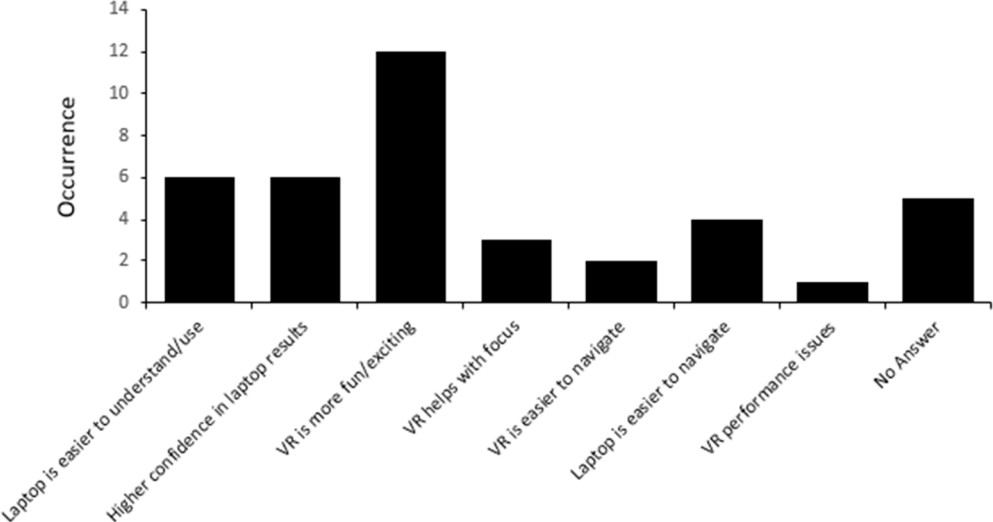

**Figure 5.** Occurrences of codes from responses to open-ended questions for all participants.

Although most participants (n = 6) preferred the reliability and confidence in results from the laptop application, the highest code occurrence was "VR is more fun/exciting" (n = 12). Participants reported being more engaged when using the immersive elements of VR, which motivated them to perform tasks correctly. Additionally, some participants mentioned having a higher level of focus when performing tasks in VR, as there were fewer outside disturbances.

### 5.3. Task Performance Similarity

Each participant performed three different tasks in both applications. In step 6b of the testing battery each participant was asked to rank different aspects of tasks on a scale from 1 to 7. This ranking focuses on the similarity of performing tasks to what it feels like to perform a similar activity at a specialist's office, where 7 stands for evaluations corresponding to situations at a professional place for checking your eyes, and 1 for the opposite, a completely unrealistic situation. Table 1 shows the average similarity scores given for each question on the laptop application, while Table 2 shows the average scores for the VR application.

**Table 1.** Similarity of task performance: Laptop.

| Task | Fixation | Smooth Pursuit | Reading |
|---|---|---|---|
| Eye tiredness | 4 | 3.85 | 3.86 |
| Move your eyes | 5 | 5.28 | 5.85 |
| Interact with environment | 4 | 4.28 | 4.57 |
| Follow instructions | 4 | 5.85 | 6.28 |

**Table 2.** Similarity of task performance: VR.

| Task | Fixation | Smooth Pursuit | Reading |
|---|---|---|---|
| Eye tiredness | 4.28 | 4.28 | 4.14 |
| Move your eyes | 4.85 | 5.28 | 5.85 |
| Interact with environment | 5.14 | 5.28 | 5.28 |
| Follow instructions | 6.14 | 5.85 | 6.42 |

Every question, except for "Move your eyes" on the fixation task, received a higher average user score in the VR application than in the laptop application. This indicates an increased sense of presence in the VR environment. While some of these aspects are not necessarily positive, such as "Eye Tiredness", their higher averages indicate that execution of screening tasks in VR produces a feeling similar to that of performing homogenous activities at a professional place for testing your eyesight.

*5.4. Participant Opinions on Using Serious Games for Vision Screening*

This subsection includes the comments from the participants to the question, "What are your opinions regarding the games?", associated with direct comments about the game experience from the interview. Table 3 shows each participant's previous experience with vision screening/testing, as well as opinions about using the developed games in VR and on laptops.

**Table 3.** Opinions on using the games to screen vision.

| P Nr. | Experience with Vision Screening/Testing | Opinions about the Used Games |
|---|---|---|
| 1 | Has some previous experience with regular vision checkups from school. | "[The VR] worked well for the things we would like to accomplish . . . Changing the size of the objects was fun." The participant also expressed that she enjoyed the fixation game more in VR for the enjoyment alone. She also mentioned appreciating the analysis application after screening on a laptop. She wished to have the possibility to screen the children's eyes alone with a trustable gamified application. |
| 2 | Wears spectacles and undergoes eye tests every second year. | "A fun way to do screening, I think. I felt that when I am performing an eye test, I focus on the games, which is good. Here the eyes may work normally, as in reality." He explained that his son is aged 5 and, as a parent, he would like to ensure that he has no vision problems. "This is an innovative way for vision testing. Although, elderly people may experience this otherwise." |
| 3 | Has regular checkups at least every second year. | The games "were well designed. Most important is to see how my vision is moving when I follow objects." "I also believe that the games are not only useful for doctors and opticians, but also for schools, universities, or other workplaces needing to measure the employees' focus or help children to learn better by showing them eventual problems with their eyes." |
| 4 | Had vision problems at a younger age, so they have experience with regular vision controls. No longer has any vision problems today. | "It is an entertaining way to test, especially in VR. I liked the easy applications. Maybe to construct an environment with more games and being able, maybe, to change the figures would be fun." "If I know the game is trustable and cost and time effective, I may test my eyes in this way rather than go to the optician. However, I believe this also has several ethical questions behind it." |
| 5 | Wears spectacles and undergoes regular controls (frequency not specified). | "I like the idea to test your eyes with games. I [also liked experiencing] more depth in 'games', which should be exploited for more traditional games as well, adding more experience." He also "believes that depth perception can be measured on laptops". |
| 6 | Wears spectacles and undergoes regular controls (frequency not specified). | "I prefer ET and games on laptop because it is faster and takes up info better. VR [was] more difficult." While it was "cooler with VR, [it was] more practical on laptops". VR was "a bit more like in reality, but it is moving from a distance". "The reading tasks was better on a laptop." It was "a bit boring to read in VR [ . . . ] you should put more interesting text there". |

**Table 3.** *Cont.*

| P Nr. | Experience with Vision Screening/Testing | Opinions about the Used Games |
|---|---|---|
| 7 | Wears spectacles and undergoes eye tests every year. | "Games easy to understand and intuitive, except that the VR had some lagging for basketball ... framerate drops. Overall, a nice experience to see and to understand how things [the eyes] are focusing ... maybe we can avoid eye specialists." |

Although their preference for either laptop or VR changed, Participants 1, 2, 4, and 5 expressed increased enjoyment when testing with games and an interest in performing future vision screenings via games. Participants 1, 3, and 7 mentioned appreciating being able to better understand their own eye movements when using the applications. The test subjects mention multiple shortcomings of the VR application, such as Participant 4 wanting more use of depth, Participant 6 finding VR more difficult to operate and the reading task boring, and Participant 7 mentioning performance issues during the smooth pursuit task.

## 6. Discussion, Limitations, and Future Work

Starting with the motivation that immersive VR has a larger FOV, and induces a higher level of presence, this paper investigates the possibilities of screening, depending on the added values, especially the higher presence [12,13]. How this presence can or cannot be correlated with a larger FOV (120 instead of 40), in general, remains to be investigated, but the possibility of experiencing large FOV and depth can be characteristics for this type of screening and is important for FVP [10]. In general, screen size, windows size, or the number of pixels utilized [57] also has to be considered [57] together with a number of environmental characteristics [56].

The sense of presence is important and is the focus of this research, and it has to be investigated more carefully, even if the tasks and technologies allowing it are not fully developed for this type of screening. The suitability and the design of the task, the utilized input–output technologies, the algorithms for integrating different sensory inputs correctly, or the utilized gamification is vital for a validated investigation. In the same way, before utilizing a medical tool, it has to be based on several different investigations with a representative population for validating it. This study is only a first step in exploring the potential of immersive VRs for screening vision problems by enumerating a few benefits and limitations. The immersiveness of virtual environments can positively influence presence, while not necessarily the task performance [58] or learning [59]. Therefore, the way we attempt to take advantage of these unique aspects of HMDs when screening FVPs has several open questions for professionals and subjects who use these technologies for screening. Some of the next questions relate to these two different stakeholders and their view on having technologies inducing high presence.

The current laptop FVP screening solution is limited by available screen size and the inability to utilize depth. Contemporary research shows that measuring depth perception is possible using HMDs [60,61]. A stereo acuity task attempting to measure depth perception in VR was developed as part of this study; however, this implementation proved difficult as it requires representation of shallow angles down to 40 s of arc [62]. A highly detailed 3D model is necessary to represent this, resulting in difficulties in measuring and lagging, that are observable for users. As there was only one developer working on the application, the implementation was abandoned. This can be further investigated in the future. It was also discovered that small errors from the eye-tracker itself lead to high variance in the gaze vector, resulting in measurements too far from the desired target. A prototype ice-hockey game was developed to test the functionality of hand-tracking technology for the purpose of performing hand–eye coordination measurements. After initial testing, the available hand-tracking proved inconsistent and unstable, making the current hardware

unfit to collect objective measurements. These limitations are discussed further in the study's related MSc thesis by Dæhlen [63], which suggests some FVP screening tasks that may be better suited for VR. This includes visual field screening, which was shown to be compatible with HMDs by Mees et al. [64], and amblyopia testing, which was proven measurable in VR by multiple research teams [36,41,65].

Presence questionnaire answers and open-ended question code analysis showed an added sense of presence when performing screening tasks in VR compared to performing similar tasks on a laptop. User experience results indicate worse user experience and less confidence in results in VR [25], which is further supported by code analysis of open-ended questions (Section 5.2) and answers to interview questions (Section 5.4). However, participants still reported increased focus and motivation despite the current issues with the application. This can be associated with the hype of VR technology.

Further analysis of gaze data is an essential next step towards OMD screening in VR. The current solution only provides superimposed eye movements over task replay and is missing graph analysis area of interest information which is present in the laptop application. Other research shows that VR can provide analysis for measuring peripheral vision [66,67]. VR can also be combined with other sensory technologies for combining measures for vision, hearing, and balance [68], measures which are correlated and affected for OMD [69].

Calibration is a necessary step when utilizing ET technologies for data collection; however, the process can reduce the immersion and engagement of users. Calibration in the different environments is reported to be similar by users; however, calibration in VR was inconsistent with those who wore glasses. This was only an issue for one out of two participants with spectacles, where they had to recalibrate multiple times to achieve sufficient gaze data quality. The issue could stem from the strength of their lenses, as gaze data quality of ET technologies in other HMDs yields lower performance for users wearing glasses [70]. Other users report having an easier time calibrating in VR, as the process is both faster and requires less strict head positioning.

A clear limitation of this study is the participant pool, with 5 out of 7 participants being in their mid-20s. As the FVP screening tools are intended to be used on school-age children, both applications should ideally have been tested on their target demographic. This is especially important when attempting to measure the added sense of presence and additional cognitive aspects that VR can bring to the vision screening domain, as participants aged from 10 to 20 tend to provide higher scores for immersion and presence [71] for this stage of the prototype. Another limitation comes from the data quality and reliability of the used HMD, which is further explained in a related publication [25]. While exact eye measurement is the heart for a successful application for FVPs, the technologies need to be revised and adjusted. As a first step, monocular calibration needs to be utilized, together with eventual other tools for better distance estimations, especially for moving in a VR environment or handling moving objects.

A major motivation for this study was to investigate the possibility of screening and rehabilitating vision training for people who need help. As we have argued, both school-age children and people after a stroke or some other brain injury would need such help. Today, there are no professionals who can perform the necessary screening and vision training for all these people. By having supporting technologies, which can complement the work of professionals or replace it, these technologies can help many. The road to this is long, but not impossible.

## 7. Conclusions

Data from both questionnaires and interviews about user experiences and presence exemplify increased interest in VR-based screening, despite the actual limitations of current technologies. Better focus and great motivation for experiences were reported when using VR, despite worse performance and lower confidence in the obtained OMD screening results. Using serious games for screening in VR was also appreciated to have great

potential, and while their role on the laptop was experienced as simplistic, opinions were certainly affected due to the hype of technology. Some users also describe being more focused when screening in VR, free from outside distractions. This highlights the higher evaluation of the sense of presence for VR in comparison with laptops, as presented in Section 5.3, with a focus on problem-solving. From the learning perspective, the more comprehensive analysis tool was appreciated, allowing users to replay and examine eye movements with more functionalities than only superimposing the eye positions on the images as we had in VR. Utilizing the possibility to screen eye movements was considered a unique aspect of VR and laptop ET technologies. The participants believed in this opportunity as a future way of screening vision, which can keep users engaged.

**Author Contributions:** Conceptualization, I.H. and A.D.; Data curation, A.D.; Investigation, A.D. and Q.A.; Methodology, I.H.; Software, A.D.; Supervision, I.H.; Validation A.D. and Q.A.; Visualization, A.D.; Writing—original draft preparation, A.D.; Writing—review and editing, A.D., I.H. and Q.A. All authors have read and agreed to the published version of the manuscript.

**Funding:** This research received no external funding.

**Data Availability Statement:** Research data collected are available in the non-published materials. These data are in the form of completed questionnaires by test participants.

**Acknowledgments:** Thanks to the SecEd project (number-267524) for allowing this paper to further develop ideas from that project. Thanks to Peter Carter for additional help during manuscript proofreading.

**Conflicts of Interest:** The authors declare no conflict of interest.

## Appendix A. Testing Battery

**Virtual Reality, Simulations, Serious Games and Eye-Tracking (XR-ET)**

**How do XR-ET technologies complement today's vision screening and training?**

A well-documented problem in vision and eye care is the lack of attention to vision diagnostics for children. Vision specialists (eye-care practitioners) are few and do not have enough resources to screen all school-aged children or help them with continuous training if needed. This screening is time- and resource-demanding and rarely includes the screening of children's functional vision [1]. The resources allocated to train the eyes, if needed, almost do not exist. The last mandatory health control at the age of 4-5 includes vision screening, while in contrast, their sight is still under development when they begin school. 25-30% of school-aged children have vision problems [2], and up to 85% of them were not detected before [3, 4]. Vision impairments can cause later impediments in academic success, education system, public health system, and social well-being [5].

The study XR-ET (hereafter, the Study) "How do XR-ET technologies complement today's vision screening and training?" aims to provide a better understanding of how XR-ETs support vision problems. The initial focus is on identifying and helping specific functional vision disturbances. The first step is to identify state-of-the-art research and development projects to increase knowledge about screening and training school children with existing technologies. The next is to test and develop XR-ET solutions to support the vision. This support cannot only come from technology development but has to be anchored with various groups involved in supporting children's health and education. The robustness, trust, and continuous availability of support for all school-aged children are essential. The long-term goal is to contribute to knowledge about integrating XR-ET technologies usable in health, schools, families, and other environments responsible for eye health.

The Study is cross-disciplinary and addresses the vision problem holistically by tackling the network of stakeholders' responsibility from children's health care (parents, school teachers, school nurses, vision care specialists) and soliciting participation for real-life demonstration of feasibility. Investigating the possibility of integrating the XR-ET technologies into social, institutional, and economic contexts to obtain a working configuration of a scalable person-centric health monitoring solution will contribute to children-centric vision care that is not existing today.

The stakeholders in the current study are *Høgskulen på Vestlandet (HVL, Bergen, Norway)*. The research project will provide significant synergies from collaboration, and we will work on applying for European research and development funds to continue. Initial contacts to build up a consortium have already been made.

For further information please contact:
- Project leader: Ilona Heldal
- Carsten Gunnar Helgesen
- Qasim Ali
- Are Dælen

[1]. Ali Q, Heldal I, Helgesen CG, Krumina G, et. Al. Current Challenges Supporting School-Aged Children with Vision Problems: A Rapid Review. Applied Sciences. 2021.
[2]. EuroSTAT, *Correction of Vision Problems by Sex, Age and Degree of Urbanization.* 2014.
[3]. Johnson, R., R. Blair, and J. Zaba, *The visual screening of title I reading students.* J Behav Optom, 2000. **11**(1).
[4]. Falkenberg, H.K., T. Langaas, and E. Svarverud, *Vision status of children aged 7–15 years referred from school vision screening in Norway during 2003–2013: A retrospective study.* BMC ophthalmology, 2019. **19**(1).
[5]. Zaba, J.N., *Children's Vision Care in the 21st Century & Its Impact on Education Literacy, Social Issues, & The Workplace: A Call to Action.* Journal of Behavioral Optometry, 2011.

Western Norway University of Applied Sciences (HVL, Bergen, Norway)

**Figure A1.** Study information.

**Virtual Reality, Simulations, Serious Games and Eye-Tracking (XR-ET)**

**Approval of participation:** How do XR-ETs complement today's vision screening and training?

Høgskulen på Vestlandet (HVL, Bergen, Norway) is conducting a research study on the use of new technologies and computer games in vision screening and training. The name of the study is "How do XR-ETs complement today's vision screening and training?" and hereafter referred to as "the Study."

The aim of this Study is to find requirements for further development and use and identify concrete practices that show under what conditions and how XR-ETs can complement today's vision screening and training. For this, several different tests with different stakeholders are planned. The results will be used for further research.

The Study is conducted in accordance with the rules set out in the Personal Data Act (1998:204). The methods used for data collection are not invasive or harmful to the participant. All personal data will be treated confidentially. Personal data will be stored in a separate file and protected with a password and login, accessible only to the project leaders. Collected results from the tests and observations, including pictures, video, and screenshots, will only be used in related research.

**About my participation**

I agree that the following conditions apply to my participation in the Study.

1. I have read and understood the information I have received about the Study.

2. I understand that my participation in the Study is voluntary, and I may choose to discontinue it at any time.

3. I accept that I am not entitled to compensation for my participation in the Study.

☐ **Yes.** I want to participate in the Study.

☐ **No.** I don't want to participate in the Study.

Phone: _____________________

Email: _____________________

Full Name:_____________________

Signature:_____________________

Date and location: _____________

**Figure A2.** Approval of participation.

**Figure A3.** Background form.

**Page 1**      **Page 2**

**Figure A4.** Usefulness and UX after screening with C&Look on a laptop.

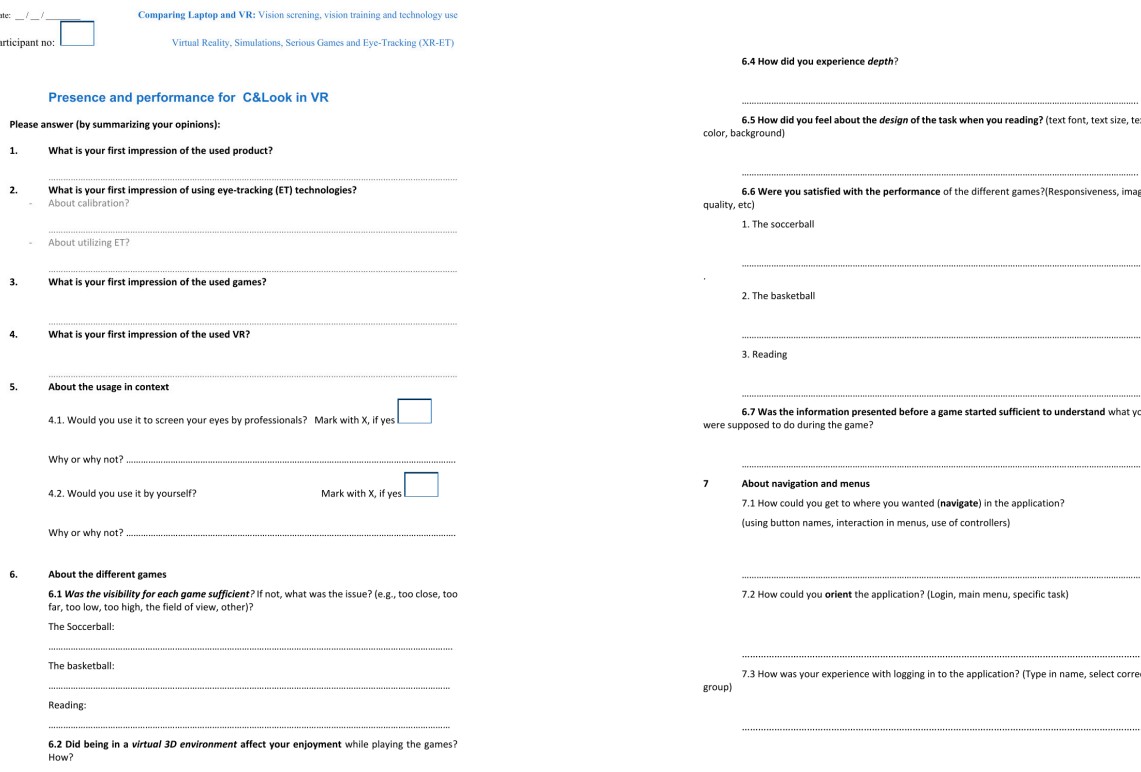

**Page 1**      **Page 2**

**Figure A5.** *Cont.*

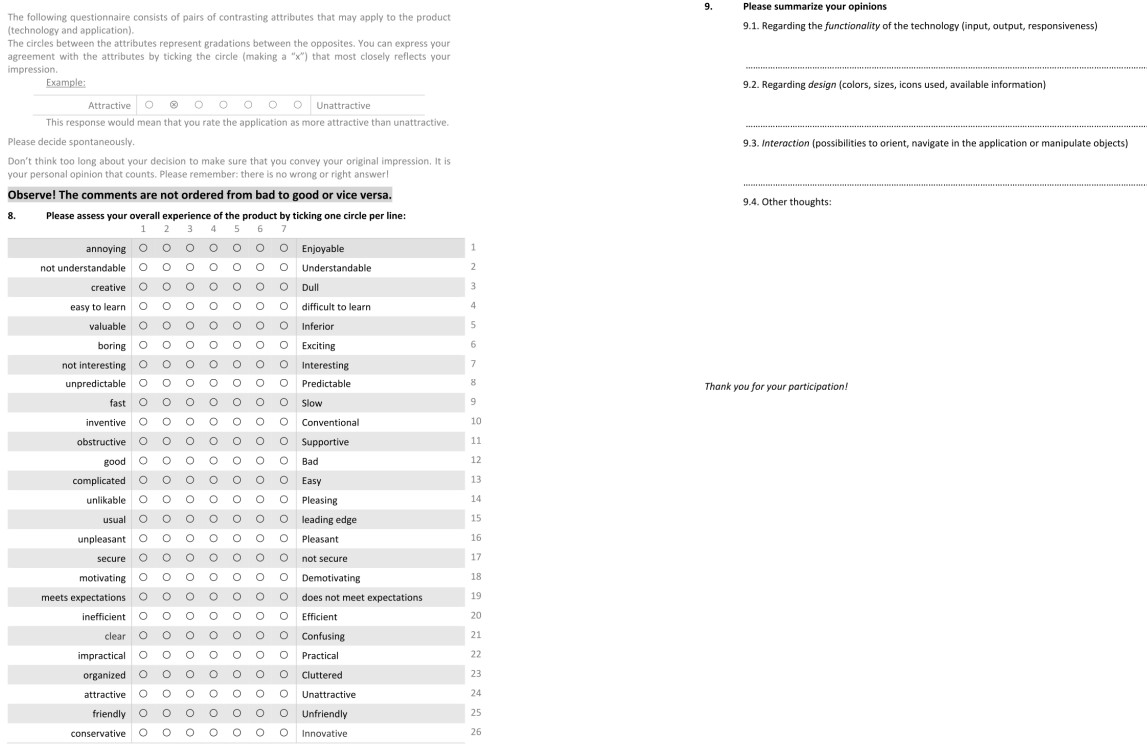

The following questionnaire consists of pairs of contrasting attributes that may apply to the product (technology and application).
The circles between the attributes represent gradations between the opposites. You can express your agreement with the attributes by ticking the circle (making a "x") that most closely reflects your impression.

Example:

Attractive ○ ⊙ ○ ○ ○ ○ ○ Unattractive

This response would mean that you rate the application as more attractive than unattractive.

Please decide spontaneously.

Don't think too long about your decision to make sure that you convey your original impression. It is your personal opinion that counts. Please remember: there is no wrong or right answer!

**Observe! The comments are not ordered from bad to good or vice versa.**

8. Please assess your overall experience of the product by ticking one circle per line:

| | 1 | 2 | 3 | 4 | 5 | 6 | 7 | | |
|---|---|---|---|---|---|---|---|---|---|
| annoying | ○ | ○ | ○ | ○ | ○ | ○ | ○ | Enjoyable | 1 |
| not understandable | ○ | ○ | ○ | ○ | ○ | ○ | ○ | Understandable | 2 |
| creative | ○ | ○ | ○ | ○ | ○ | ○ | ○ | Dull | 3 |
| easy to learn | ○ | ○ | ○ | ○ | ○ | ○ | ○ | difficult to learn | 4 |
| valuable | ○ | ○ | ○ | ○ | ○ | ○ | ○ | Inferior | 5 |
| boring | ○ | ○ | ○ | ○ | ○ | ○ | ○ | Exciting | 6 |
| not interesting | ○ | ○ | ○ | ○ | ○ | ○ | ○ | Interesting | 7 |
| unpredictable | ○ | ○ | ○ | ○ | ○ | ○ | ○ | Predictable | 8 |
| fast | ○ | ○ | ○ | ○ | ○ | ○ | ○ | Slow | 9 |
| inventive | ○ | ○ | ○ | ○ | ○ | ○ | ○ | Conventional | 10 |
| obstructive | ○ | ○ | ○ | ○ | ○ | ○ | ○ | Supportive | 11 |
| good | ○ | ○ | ○ | ○ | ○ | ○ | ○ | Bad | 12 |
| complicated | ○ | ○ | ○ | ○ | ○ | ○ | ○ | Easy | 13 |
| unlikable | ○ | ○ | ○ | ○ | ○ | ○ | ○ | Pleasing | 14 |
| usual | ○ | ○ | ○ | ○ | ○ | ○ | ○ | leading edge | 15 |
| unpleasant | ○ | ○ | ○ | ○ | ○ | ○ | ○ | Pleasant | 16 |
| secure | ○ | ○ | ○ | ○ | ○ | ○ | ○ | not secure | 17 |
| motivating | ○ | ○ | ○ | ○ | ○ | ○ | ○ | Demotivating | 18 |
| meets expectations | ○ | ○ | ○ | ○ | ○ | ○ | ○ | does not meet expectations | 19 |
| inefficient | ○ | ○ | ○ | ○ | ○ | ○ | ○ | Efficient | 20 |
| clear | ○ | ○ | ○ | ○ | ○ | ○ | ○ | Confusing | 21 |
| impractical | ○ | ○ | ○ | ○ | ○ | ○ | ○ | Practical | 22 |
| organized | ○ | ○ | ○ | ○ | ○ | ○ | ○ | Cluttered | 23 |
| attractive | ○ | ○ | ○ | ○ | ○ | ○ | ○ | Unattractive | 24 |
| friendly | ○ | ○ | ○ | ○ | ○ | ○ | ○ | Unfriendly | 25 |
| conservative | ○ | ○ | ○ | ○ | ○ | ○ | ○ | Innovative | 26 |

9. **Please summarize your opinions**

9.1. Regarding the *functionality* of the technology (input, output, responsiveness)

..................................................................................................................................................................

9.2. Regarding *design* (colors, sizes, icons used, available information)

..................................................................................................................................................................

9.3. *Interaction* (possibilities to orient, navigate in the application or manipulate objects)

..................................................................................................................................................................

9.4. Other thoughts:

*Thank you for your participation!*

**Page 3** **Page 4**

**Figure A5.** Usefulness and UX after screening with C&Look in VR.

Date: __ / __ /____  **Comparing Laptop and VR:** Vision screening, vision training and technology use

Participant no: ____  Virtual Reality, Simulations, Serious Games and Eye-Tracking (XR-ET)

**Presence and performance for comparing C&Look on Laptop and in VR**

Please think back to performing the tasks on the Laptop and VR.

1. Comparing with experiences while checking your eyes at a physical place, e.g., at an optician or a doctor's office, can you argue why (or why not) you would like to use a similar application on
   a. Laptop
   ...............................................................................................................................
   b. VR
   ...............................................................................................................................

2. In which application did you find it easier to navigate? Laptop or VR? Why?
   ...............................................................................................................................

3. Compared to performing the tasks on a laptop, did the addition of depth in VR change your enjoyment/immersion? Why or why not?
   ...............................................................................................................................

4. Were there any features from either application you felt were lacking from the other? If so, what?
   ...............................................................................................................................

5. Please rate your overall experiences on calibration in each environment.

| | **Laptop** | | | | | | | **VR** | | | | | | |
|---|---|---|---|---|---|---|---|---|---|---|---|---|---|---|
| | Difficult | | | | | Very Easy | | Difficult | | | | | Very Easy | |
| | 1 | 2 | 3 | 4 | 5 | 6 | 7 | 1 | 2 | 3 | 4 | 5 | 6 | 7 |
| Calibration | ☐ | ☐ | ☐ | ☐ | ☐ | ☐ | ☐ | ☐ | ☐ | ☐ | ☐ | ☐ | ☐ | ☐ |

Comments:
...............................................................................................................................

6. Rate your **experiences** of using Laptop and VR on a scale of 1 to 7

7 stands for evaluations corresponding to situations at a place for checking your eyes, being tested by experts and 1 for the opposite, a completely unrealistic situation.

| | **Laptop** | | | | | | | **VR** | | | | | | |
|---|---|---|---|---|---|---|---|---|---|---|---|---|---|---|
| | Low 1 | 2 | 3 | 4 | 5 | 6 | High 7 | Low 1 | 2 | 3 | 4 | 5 | 6 | High 7 |
| experiences | ☐ | ☐ | ☐ | ☐ | ☐ | ☐ | | ☐ | ☐ | ☐ | ☐ | ☐ | ☐ | |
| engagement | ☐ | ☐ | ☐ | ☐ | ☐ | ☐ | | ☐ | ☐ | ☐ | ☐ | ☐ | ☐ | |
| involvement | ☐ | ☐ | ☐ | ☐ | ☐ | ☐ | | ☐ | ☐ | ☐ | ☐ | ☐ | ☐ | |
| focus of attention | ☐ | ☐ | ☐ | ☐ | ☐ | ☐ | | ☐ | ☐ | ☐ | ☐ | ☐ | ☐ | |
| experiencing time | ☐ | ☐ | ☐ | ☐ | ☐ | ☐ | | ☐ | ☐ | ☐ | ☐ | ☐ | ☐ | |
| Compared to real memories | ☐ | ☐ | ☐ | ☐ | ☐ | ☐ | | ☐ | ☐ | ☐ | ☐ | ☐ | ☐ | |

7. Rate your **performance** for each task for Laptop and VR on a scale of 1 to 7.
   7 stands for your evaluations corresponding to a performance at a place for checking your eyes, when being tested by experts. 1 stands for the opposite, a very low performance.

| | **Laptop** | | | | | | | **VR** | | | | | | |
|---|---|---|---|---|---|---|---|---|---|---|---|---|---|---|
| | Low 1 | 2 | 3 | 4 | 5 | 6 | High 7 | Low 1 | 2 | 3 | 4 | 5 | 6 | High 7 |
| **Soccerball** | | | | | | | | | | | | | | |
| Eye tiredness | ☐ | ☐ | ☐ | ☐ | ☐ | ☐ | | ☐ | ☐ | ☐ | ☐ | ☐ | ☐ | |
| Move your eyes | ☐ | ☐ | ☐ | ☐ | ☐ | ☐ | | ☐ | ☐ | ☐ | ☐ | ☐ | ☐ | |
| Interact with environment | ☐ | ☐ | ☐ | ☐ | ☐ | ☐ | | ☐ | ☐ | ☐ | ☐ | ☐ | ☐ | |
| Follow instructions | ☐ | ☐ | ☐ | ☐ | ☐ | ☐ | | ☐ | ☐ | ☐ | ☐ | ☐ | ☐ | |
| **Basketball** | | | | | | | | | | | | | | |
| Eye tiredness | ☐ | ☐ | ☐ | ☐ | ☐ | ☐ | | ☐ | ☐ | ☐ | ☐ | ☐ | ☐ | |
| Move your eyes | ☐ | ☐ | ☐ | ☐ | ☐ | ☐ | | ☐ | ☐ | ☐ | ☐ | ☐ | ☐ | |
| Interact with environment | ☐ | ☐ | ☐ | ☐ | ☐ | ☐ | | ☐ | ☐ | ☐ | ☐ | ☐ | ☐ | |
| Follow instructions | ☐ | ☐ | ☐ | ☐ | ☐ | ☐ | | ☐ | ☐ | ☐ | ☐ | ☐ | ☐ | |
| | **Laptop** | | | | | | | **VR** | | | | | | |

**Page 1** **Page 2**

**Figure A6.** *Cont.*

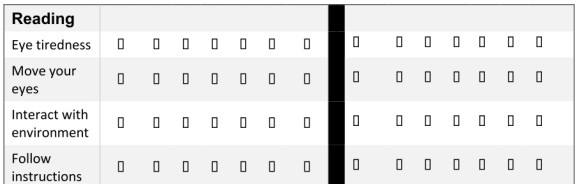

*Thank you for your participation!*

**Page 3**

**Figure A6.** Presence and performance for comparing C&Look on a laptop and in VR.

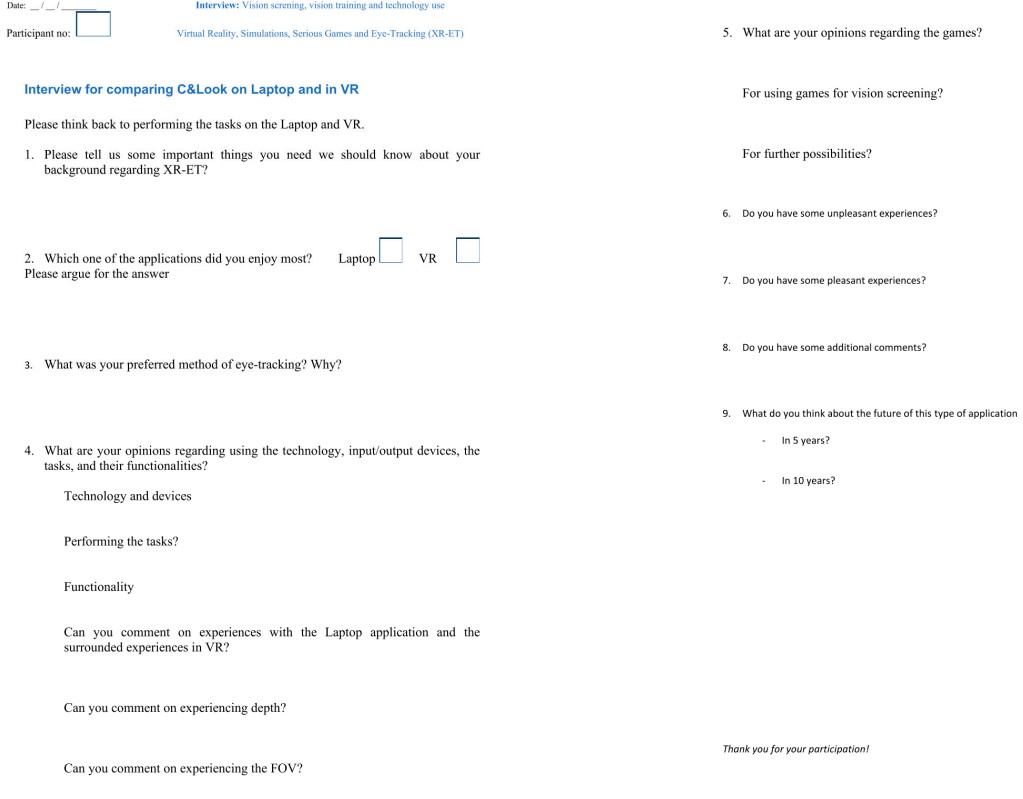

**Page 1**       **Page 2**

**Figure A7.** Interview questions for comparing C&Look on a laptop and in VR.

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
