# Peer review of "Technologies Supporting Screening Oculomotor Problems: Challenges for Virtual Reality"

_computers, doi:10.3390/computers12070134_

Round 1

Reviewer 1 Report

Dear Authors!
The work is interesting and certainly useful, both from a theoretical and practical point of view, both for specialists in the development of applied software (complexes and solutions) based on engines supporting virtual and / or augmented reality technologies, and for specialists in the field of functional diagnostics and screening vision examinations.
And despite the overall positive impression, there are a number of points that you should pay your attention to before further promotion of the manuscript, namely:
1. An immersive approach in medical diagnostics is defined as a set of techniques and methods for organizing the interaction of process participants in a virtual test environment that provides interactivity of diagnostics due to sensory multivector impact on a patient (subject). How is your approach fundamentally different from the existing immersive approaches in medical diagnostics implemented using gaming artificial intelligence, for example, based on the Unreal Engine 4.5 or 5?
2. In your work, give a more detailed description of the barriers to the implementation of immersive technologies regarding your proposals.
3. The paper does not note the possibility of a discrepancy between the actual level of objective and subjective assessment.
4. Add in the second section a brief overview of the frequency and structure of visual anomalies - this will cause more interest among specialists / engineers in the field of development of biotechnical devices and systems.
5. In section 3.2, your contribution is very modestly disclosed - describe your development in more detail (in my opinion, this is the most valuable thing in your work!!!).
6. From the point of view of a systematic approach, it is required to respect the structural and functional organization of the proposed solutions. Do not limit yourself to information and software. I strongly recommend adding this to your work.
7. Analysis of information allows you to evaluate the state of visual functions in dynamics. Give recommendations at the end of the work to improve the efficiency of the end users of your ideas, for example, when carrying out treatment and preventive measures.
8. The Discussion Section should estimate approaches, described in the Introduction Section.
Good luck with revisions!

It is required to re-subtract the style and agree on the end of sentences.

Author Response

Thank you for all the great comments! Responses to the comments are in the attached file.

Reviewer 2 Report

The research question addressed is quite interesting. The study done needs to be more detailed and richer. Specifically focusing on the factors that might impact the user experience with the VR HMDs. There is a potential in using VR HMDs but, to what extent we are really ready to opt this technology for Vison screening since it does impact already the vison and the brain. The questions used need to be reformulated to capture more details about the user experience. In order to have a complete study, the survey answers should not be the only element used to measure the results.

Minor 

Author Response

Responses to the comments are in the attached file. I cant upload the revised manuscript before these answers have been reviewed according to the website.

Round 2

Reviewer 1 Report

Dear Authors!

All comments were taken into account.

Personally, I have no more questions on this article.

I wish you success.

Some bugs have been fixed.

However, I still think that the work should be proofread by a native speaker.